# Deep Learning for Automated Detection and Identification of Migrating American Eel *Anguilla rostrata* from Imaging Sonar Data

Xiaoqin Zang [1], Tianzhixi Yin [2], Zhangshuan Hou [1], Robert P. Mueller [1], Zhiqun Daniel Deng [1] and Paul T. Jacobson [3,*]

1 Energy and Environment Directorate, Pacific Northwest National Laboratory, Richland, WA 99354, USA; xiaoqin.zang@pnnl.gov (X.Z.); zhangshuan.hou@pnnl.gov (Z.H.); robert.mueller@pnnl.gov (R.P.M.); zhiqun.deng@pnnl.gov (Z.D.D.)
2 National Security Directorate, Pacific Northwest National Laboratory, Richland, WA 99354, USA; tianzhixi.yin@pnnl.gov
3 Electric Power Research Institute, Palo Alto, CA 94304, USA
* Correspondence: pjacobson@epri.com; Tel.: +1-(443)-280-0294

**Abstract:** Adult American eels (*Anguilla rostrata*) are vulnerable to hydropower turbine mortality during outmigration from growth habitat in inland waters to the ocean where they spawn. Imaging sonar is a reliable and proven technology for monitoring of fish passage and migration; however, there is no efficient automated method for eel detection. We designed a deep learning model for automated detection of adult American eels from sonar data. The method employs convolution neural network (CNN) to distinguish between 14 images of eels and non-eel objects. Prior to image classification with CNN, background subtraction and wavelet denoising were applied to enhance sonar images. The CNN model was first trained and tested on data obtained from a laboratory experiment, which yielded overall accuracies of >98% for image-based classification. Then, the model was trained and tested on field data that were obtained near the Iroquois Dam located on the St. Lawrence River; the accuracy achieved was commensurate with that of human experts.

**Keywords:** American eel; imaging sonar; image classification; fish identification; deep learning; convolutional neural network

## 1. Introduction

The American Eel (*Anguilla rostrata*) is the only member of the genus *Anguilla* found in North America. It spends most of its life in fresh and estuarine waters and migrates to the Sargasso Sea in the Atlantic Ocean to spawn. Historically abundant throughout lakes, streams, and rivers flowing into the Atlantic Ocean, including the St. Lawrence River and Lake Ontario, the American eel has suffered from a dramatic population decline [1–6]. From the early 1980s to 2006, the abundance of eels in these waters decreased by almost 90% [1–3], which makes the American eel a species of management and regulatory concern. The species was listed as Endangered by the province of Ontario (Canada) in 2007 [3]. The Atlantic States Marine Fisheries Commission declared the American eel depleted in U.S. waters [4]. The species is listed as Endangered over its entire range by the International Union for the Conservation of Nature (IUCN) [5]. An even more dramatic decline in the abundance of the closely-related European eel (*Anguilla anguilla*) has occurred over a similar period [6], and the species is listed as Critically Endangered by the IUCN. The dramatic decline of these eel populations is attributed to several factors including the construction and operation of hydroelectric facilities, degradation and pollution of habitats, commercial harvest, and changes in ocean-atmospheric conditions affecting their marine life stages [7]. Hydroelectric facilities affect the species in at least two ways: (1) the dams

impede upstream migration of juveniles [8], and (2) adult migrants are exposed to risk of injury and mortality when they pass downstream via hydroelectric turbines [9,10].

Design, operation, and optimization of downstream passage facilities that are economically and biologically effective requires knowledge of eel behavior during their downstream migration including when eels are approaching hydropower facilities, their pathways of approach and passage, and their near-field behavioral responses to facility structures such as intakes, guidance structures, and bypasses. However, such information on how outmigrating eels interact with dams is largely unknown. Moreover, the outmigration of adult eels is episodic and protracted, typically extending over a period of several months each year [11]. Since technologies are lacking to automatically detect migrating eels, dam operators may be required to curtail turbine operation and spill water at night throughout the outmigration season to ensure safe eel passage. Given the protracted nature of adult eel outmigration, cost-effective monitoring of eel passage requires a high degree of automation for data processing.

Imaging sonar technology is an effective way to monitor fish passage in turbid and dark environments, which enables all-day real-time monitoring of eels near hydropower facilities. The Electric Power Research Institute (EPRI) has demonstrated the feasibility of monitoring eel migration with imaging sonars near hydropower facilities [12]. The study found that adaptive resolution imaging sonar (ARIS) was capable of correctly identifying eels at a range of up to 20 m. In other previous studies, ARIS and DIDSON (Dual frequency IDentification SONar), which is the older version of ARIS, have been used to document the behavior of downstream and upstream migrating fish at dams and to enumerate migrating fish in streams and rivers [13–15]. Widespread use of DIDSON and ARIS has provided a large amount of surveillance data. An ability to process these data efficiently and develop a real-time automatic identification method is critically important to the design and operation of eel passage facilities at hydropower dams.

The first attempt to develop computer-driven eel identification from DIDSON images was made by Mueller et al. [16], who tested a simple neural network model on a small dataset (57 eel images and 130 debris images) and achieved a 5% false positive rate and a 7% false negative rate. In addition, Bothmann et al. [17] performed real-time classification of eel and trout in DIDSON video sequences using several machine learning algorithms including support vector machines and random forests. The two studies successfully demonstrated the feasibility of computer-driven eel identification from sonar images; however, since both studies were performed with low-noise short-range (5 to 10 m) datasets, the challenge of eel identification in noisy field data at detection ranges over 10 m is still unresolved.

Deep learning methods have been used in many environmental and ecological research areas such as animal species identification [18], ambient air pollution prediction [19], and automatic land cover classification [20]. The convolutional neural network (CNN), one of the most popular deep learning structures, is a powerful method for image classification and object detection [21–25]. It outperforms traditional feature-based machine learning approaches thanks to the advantages of [26]: (1) Automatic feature extraction: the convolutional layers can learn features automatically by striding filters through the image without manual feature-engineering; and (2) hierarchical feature extraction (CNN can learn features from the data at different levels including small details, complex patterns, and the big picture). In recent years, CNN has become the leading machine learning model for image classification and video analysis in the fields of transportation, medicine, finance, and security [27]. In fisheries research, CNN has been applied to fish recognition in underwater optical camera footage [28].

The great success of CNN in optical image classification indicates its potential for sonar image classification, except that sonar images might be far more difficult than optical images to classify. The challenge of sonar image classification includes: (1) sonar images are pseudo images formed by sound reflections in the direction of the beam axis of acoustic transducers (i.e., the orientation of the object relative to the beam axis of transducers

significantly affects the apparent shape of the object in sonar images); (2) sonar images are contaminated by unavoidable noise induced by the ambient environment; and (3) sonar images have much lower resolution than optical images due to physical and hardware limitations, which makes the classification more difficult.

Our goal is to identify American eel with CNN from sonar images. The study was performed with data from a controlled laboratory experiment and existing data from previous field experiments [12]. Background removal and wavelet denoising were employed to enhance sonar images before CNN classification. We successfully identified eels from sonar images at a range of 35 m in the field. The designed data analysis framework will enable the automation of fish identification, enumeration, and behavior monitoring for hydropower infrastructure design, operation, and optimization. This method can be further applied to the identification of other fish species such as Sea lamprey (*Petromyzon marinus*) and Pacific lamprey (*Entosphenus tridentatus*), which have a body shape and swimming behavior comparable to eel.

## 2. Materials and Methods

### 2.1. Data Collection

#### 2.1.1. Laboratory Experiments

Due to the scarcity of labeled field data and the high-level of environmental noise in natural rivers, we designed a controlled environment to record the swimming of American eels and the motion of non-eel objects that could cause a high false positive classification rate. The objective of the laboratory experiment was to collect high-quality sonar data in a well-controlled laboratory experiment and to test the validity of the proposed data analytics method. Four American eels (length: 330–335 mm; diameter: 20–24 mm) were tested. The non-eel objects were two neutrally buoyant wood sticks with similar dimensions. The smaller stick was 317 mm in length and 19 mm in diameter. The larger stick was 600 mm in length and its diameter ranged from 15 to 25 mm. Both eels and sticks were tethered (Figure S1), with handling details included in the Supplementary Materials.

Tests were conducted at the Aquatic Research Laboratory at the Pacific Northwest National Laboratory (PNNL) in an oval-shaped fiberglass test tank (7.3 m long, 3.0 m wide, and 2.5 m deep) (Figure 1a). The water was pumped from the adjacent Columbia River and the temperature ranged from 10.5 to 11.0 °C. Two water flow speeds were tested (0.53 and 0.76 m/s) and each eel was swimming for a total of three to five minutes at each flow speed. To create the two flow speeds, a stacked array of twenty independent submersible pumps (Pentair Flotech, Model FP0S3000X, London, UK) in a 4 × 5 grid (Figure 1b) were mounted in an aluminum frame, which rested on a raised platform (Figure S2). Details of the setup of the pump array and the generation of the flow field are included in the Supplementary Materials. To reduce reflections from the pump array and platform, two sections of 122 × 61 × 2.65 cm (length × width × thickness) anechoic material (Precision Acoustics AptFlex F48, Precision Acoustics Ltd., Dorchester, Dorset, UK) were used, which can absorb sound in the 10 kHz to 1.5 MHz band.

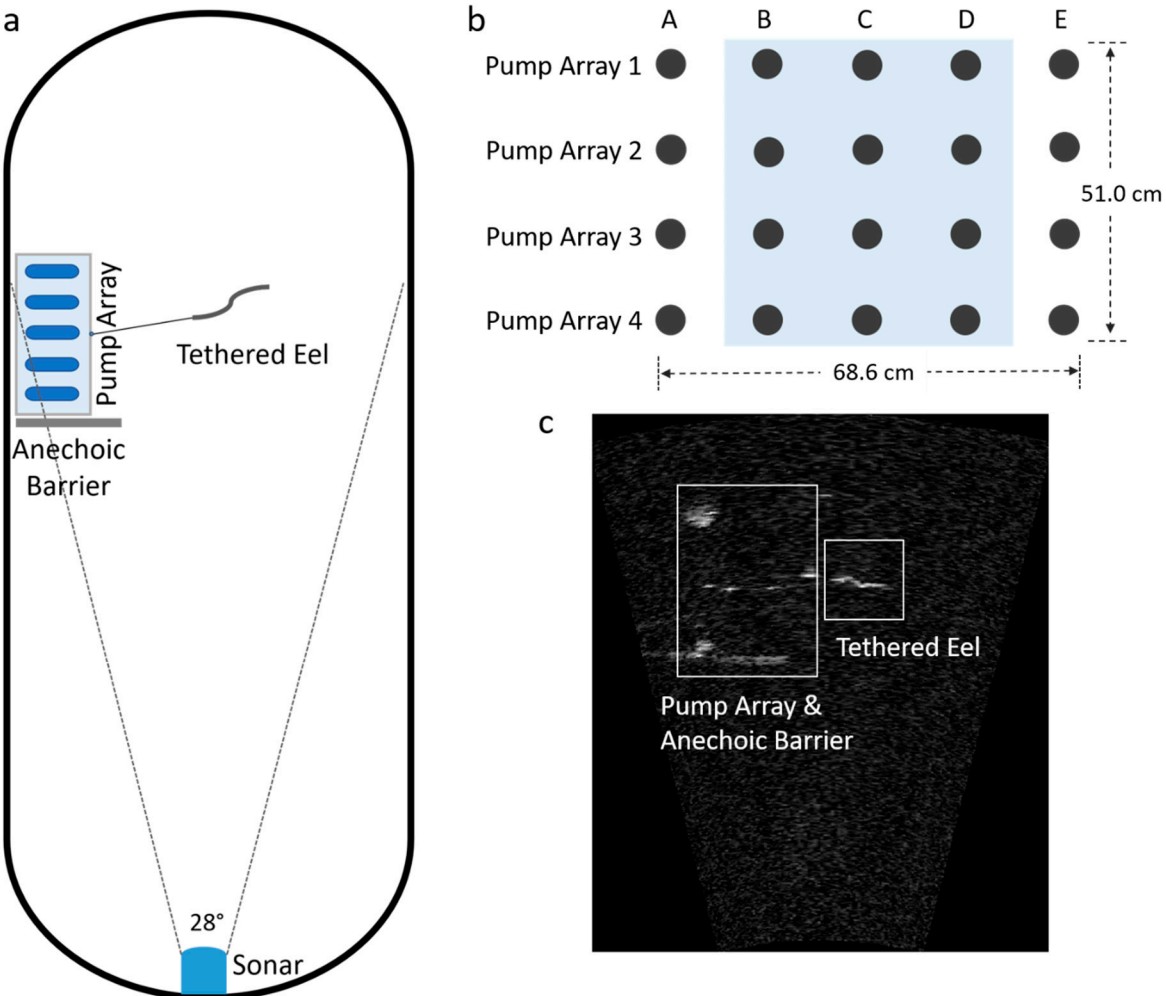

**Figure 1.** The setup of the laboratory experiments. (**a**) Test tank component diagram including the oval-shaped tank (7.3 m long, 3.0 m wide, and 2.5 m deep), the pump array, the anechoic barrier, sonar, and the tethered eel. (**b**) Diagram of pump arrangement used to generate flow field in the test tank. Shaded region indicates the region where velocity measurements were made and region where eels were confined to be imaged. (**c**) An example of the sonar image snapshot showing a tethered eel at a 5-6 m detection range and the pump array and anechoic barrier in the background.

Eel swimming behavior and neutrally buoyant stick motion were recorded with an ARIS Explorer 1800 sonar (Sound Metrics Corp., Bellevue, WA, USA) mounted at mid-water depth at the opposite end of the tank (Figure 1a). The ARIScope software version 2.6.2 (Sound Metrics Corp., Bellevue, WA, USA) was used to control the sonar and record data (see sonar settings in Table 1). Eels were tethered so that they could swim within the central portion of the sonar beams (e.g., Figure 1a,c).

An underwater optical camera (LED Multi SeaCam Model 2065, Deep Sea Power and Light, San Diego, CA, USA) was positioned at the water surface looking down to the tank (Figure S2) to monitor the motion of eels and sticks. The optical camera was connected to a digital video recorder, which was synchronized to the sonar video files and used to verify periods of normal eel swimming behavior (anguilliform/sinusoidal locomotion).

**Table 1.** The experimental setup and the settings of the ARIS sonar.

| Parameters | Values |
| --- | --- |
| Flow speed in the fish swimming zone | High flow: 0.76 m/s; Low flow: 0.53 m/s |
| Range from the sonar to the fish swimming zone | 5.5 m |
| Detection range | 2.8–6.7 m |
| Focus range | 5.7 m |
| Operating frequency | 1.1 MHz |
| Number of beams | 96 |
| Number of samples per beam | 537 or 482 |
| Resolution | 5.8 mm or 7.3 mm |

### 2.1.2. Field Experiments

Field data were collected as part of a sonar technology evaluation project at the Iroquois Water Control Dam on the St. Lawrence River in 2015 [12], with the experimental site and setup shown in Figure S3 in the Supplementary Materials. Selected acoustic targets were tethered to surface floats with monofilament fishing line, including adult eels (720–910 mm long), PVC pipes and waterlogged sticks of similar dimensions, a large eel-type fishing lure, and other species of live fish. These acoustic targets were passed through the sonar beams at various distances and depths. Additionally, groups of untethered eels were released immediately upstream of the sonar, and macrophyte mats were documented passing through the acoustic beams via direct visual observation. EPRI (2017) documented three sonar systems (details in the Supplementary Materials) that can be used to track known eels; however, only ARIS can be used to identify eels based upon acoustic data alone, and thus the ARIS data were utilized in this study.

### 2.2. Data Analysis

#### 2.2.1. Data Cleaning and Labeling

In the laboratory experiment, the tethered eels were swimming against the flow generated by the pump array, which sometimes affected their body control and natural swimming pattern. During some tests using the highest water flow speed, eels would sometimes lose body control and their body would be dragged by the flow like a straight stick. Thus, prior to object detection, the time periods during which eels had active body control were carefully selected based on the recordings of the overhead optical camera. Moreover, moments when eels were surrounded with entrained air due to sporadic, uneven, or turbulent flow emanating from the pump array were excluded before applying the object detection algorithm. The cleaned sonar video clips were labeled as eel or stick according to the experimental log.

In the field data, sonar records encompassing tethered eels and tethered sticks and PVC pipes were identified by cross-referencing the study reports and sonar metadata. Sonar records for the acoustic targets, marked as stick or PVC pipe, but not clearly linked to a tethered release, were identified based on visual examination of the sonar record (i.e., professional judgement). This visual examination included repeated playback with manipulation of display settings such as playback speed and upper and lower intensity thresholds. The identified records were trimmed to comprise the contiguous series of frames during which the known target was within the acoustic field, or 100 contiguous frames encompassing such a series, whichever was longer.

#### 2.2.2. Sonar Image Processing

The raw data in .aris format were converted to grayscale regular-grid images with an open source MATLAB script (see https://github.com/nilsolav/ARISreader, accessed on 06/30/2021) [29]. The data conversion corrected lens distortion due to sonar beam configuration. An example image after data conversion is shown in Figure 2a, in which the sonar detection range was 2.0–17.8 m in the field test. The number of beams was 48, the

operating frequency was 1.1 MHz, and the image size was 512 × 944 pixels. The image captured a 76-cm-long adult eel, located approximately 6 m from the sonar.

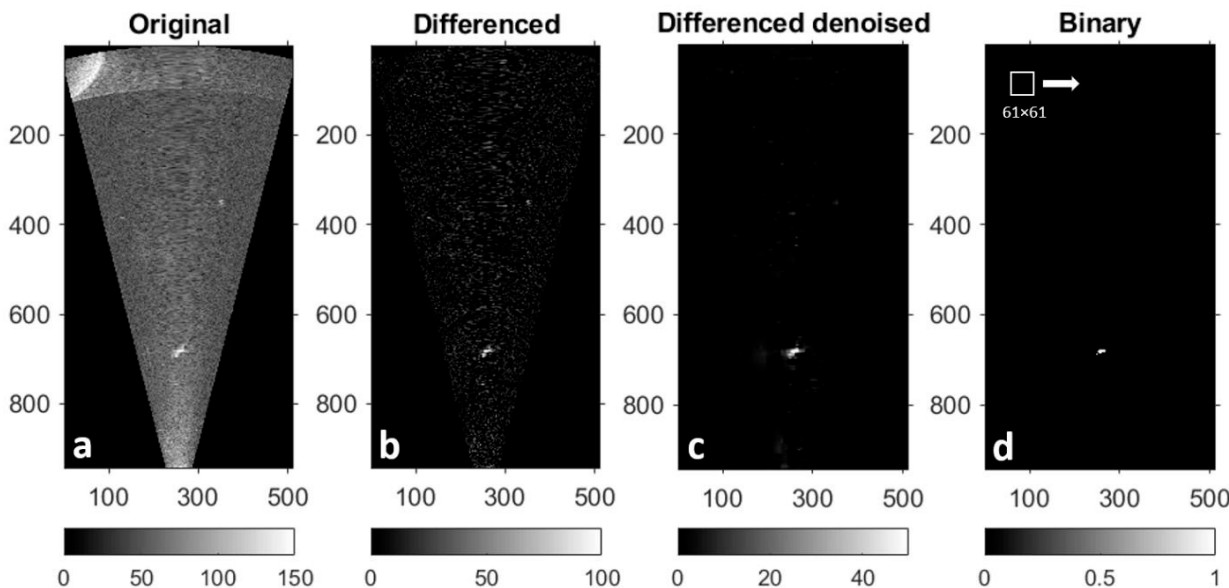

**Figure 2.** The workflow of sonar image processing and object extraction. (**a**) The original image in regular-grid was converted from the .aris file. (**b**) The differenced image was obtained by subtracting the mean of four adjacent frames from the current frame. (**c**) The differenced image was denoised using a wavelet transform. (**d**) The denoised image was transformed into a binary image with a threshold in pixel intensity. A sliding window of 61 × 61 pixels moved from the upper left to the bottom right of the binary image to screen potential objects of interest. The horizontal and vertical axes denote the pixel numbers of each image. Grayscale bars denote the pixel intensities.

Sonar images may contain not only the objects for detection, but also static structures in the background. In the laboratory experiment, the background of sonar images was mainly the pump array and the anechoic barrier (e.g., Figure 1c). In the field data, the background in several cases was the dam structures (e.g., Figure 2a). In addition to major background objects, sonar images contain ambient noise characterized by small size, random location, and high pixel intensity. The noise in sonar images comes from either ambient environmental noise that occurs in a similar operating frequency of the sonar, or scattering of sonar signals by entrained air or small debris. Since background structures are static and consistent throughout the data file, they can be removed through image subtraction. Figure 2b shows an image that was subtracted by the mean of four adjacent images. After image subtraction, the dam structure on the upper left corner was removed. The image subtraction also removed part of the background noise.

Wavelet denoising is a classical approach to enhancing images corrupted by Gaussian noise. The 2D wavelet denoising involves computation of the wavelet transform represented by several orthonormal detail coefficients and one approximate component co-efficient. The detail components are filtered to reduce noise, followed by inverse transformation to reconstruct data from which noise has been filtered [30–32]. Various techniques can be used for 2D image enhancement, digital filtration, and feature identification, which are usually based on sinusoidal basis functions [33,34]. Discrete wavelet transform using Haar or Daubechies wavelets [35,36] is among the most promising approaches and has been popular in image coding, edge extraction, and binary logic design. In this study, multiple types of wavelets with various threshold levels were tested to select the wavelet that could efficiently remove noise from images while maintaining important edges of the object. In comparison, the Daubechies 2 wavelet was selected for image enhancement. Through wavelet transform, filtering, and reconstruction, the small speckles in sonar images were removed, with an example shown in Figure 2c.

### 2.2.3. Object Extraction

After background removal and wavelet denoising, the sonar data were processed to extract images of eels or sticks/PVC pipes for training and testing the CNN model. Since only one object of interest was recorded in each video clip after data cleaning, a straightforward object detection algorithm was designed as follows (Figure 2d). (1) A threshold of pixel intensity was selected, and the grayscale image was transformed into a binary image by setting the pixel intensities above the threshold to 1 and the pixel intensities below the threshold to 0. (2) A sliding window of $61 \times 61$ pixels moved from the upper left to bottom right, screening the potential object with a threshold of the number of white pixels (where pixel intensities are equal to 1). Once the number of white pixels in the sliding window reached its maximum and exceeded the threshold, the current location of the sliding window was recorded for object extraction. (3) The extracted objects were visually examined to ensure that they were the intended targets. Representative examples of the extracted images of eels and sticks from the laboratory data and field data are shown in Figure 3 (after background removal) and Figure 4 (after background removal and wavelet denoising), respectively. The $61 \times 61$-pixel window perfectly accommodated the eel and non-eel objects in both the laboratory and the field data.

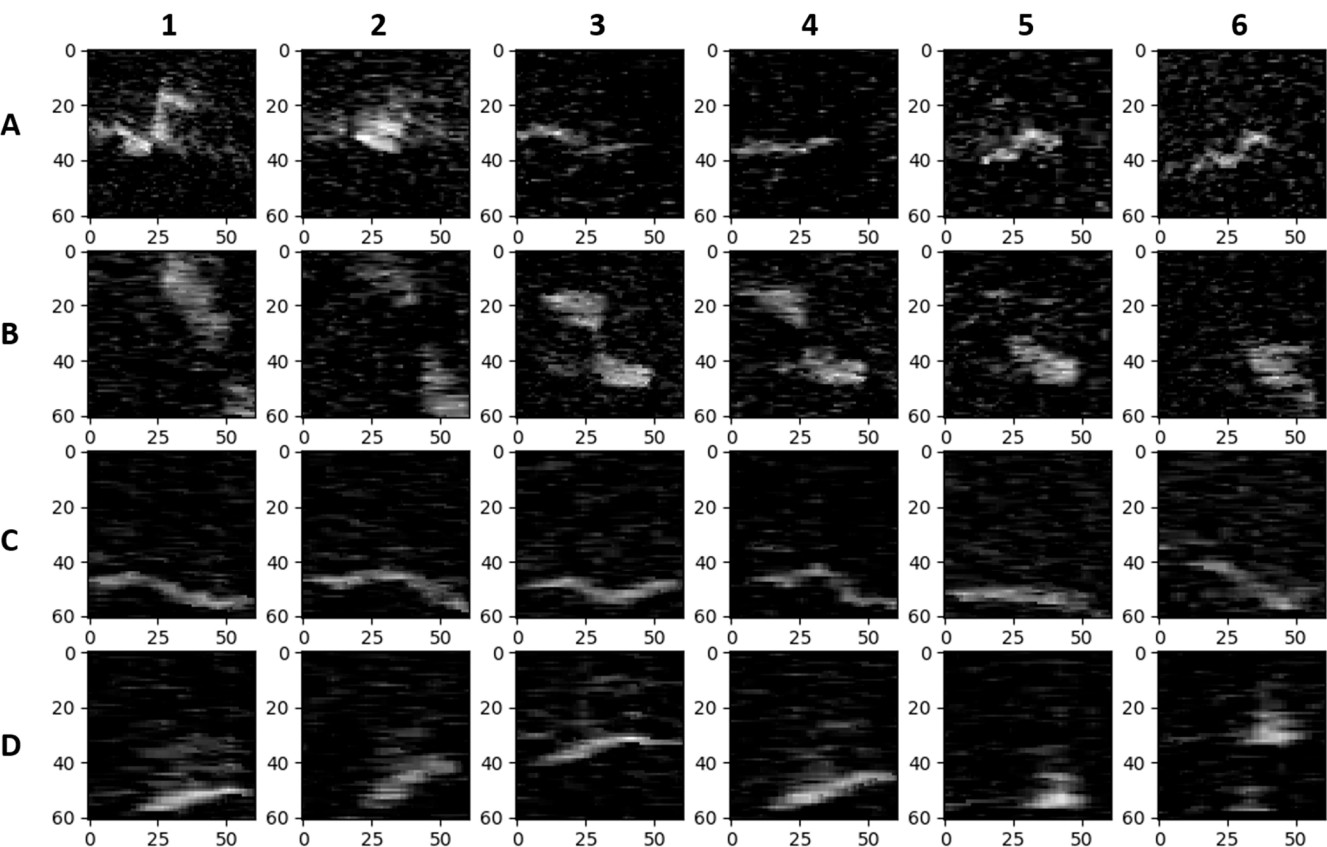

**Figure 3.** Example images ($61 \times 61$ pixels) of eels and sticks after background removal: eels in the field (Row **A**), sticks in the field (Row **B**), eels in the laboratory (Row **C**), and sticks in the laboratory (Row **D**). Column 1 and 2 are the same target at different times, the same for columns 3 and 4 as well as columns 5 and 6.

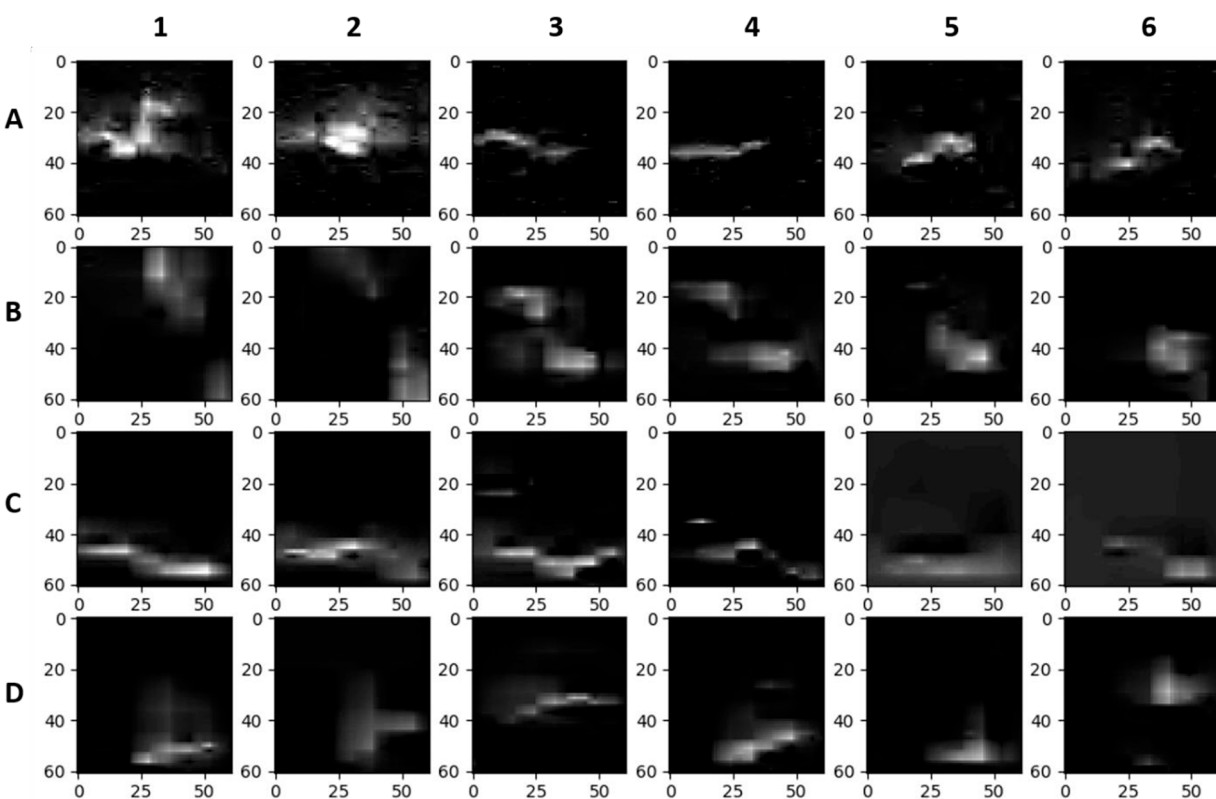

**Figure 4.** Example images (61 × 61 pixels) of eels and sticks after background removal and wavelet denoising: eels in the field (Row **A**), sticks in the field (Row **B**), eels in the laboratory (Row **C**), and sticks in the laboratory (Row **D**). Columns 1 and 2 are the same target at different times, the same for columns 3 and 4 as well as columns 5 and 6.

2.2.4. Statistical Analysis of Sonar Images: Aspect Ratio and Orientation Angle of Objects

The aspect ratio and orientation angle of the detected objects were calculated to differentiate eels from sticks. Since eels have a more flexible body than sticks, clear differences were expected in the aspect ratio and orientation angle of their images. First, the major axis of the object in the binary image, which was denoted by the two white pixels with the longest distance among all white pixel pairs, was found through a brute-force search. Then, the aspect ratio of the object was calculated by

$$r = \frac{\delta y^2 + \delta x^2}{S}, \tag{1}$$

where $\delta y = y_2 - y_1$, $\delta x = x_2 - x_1$ are the vertical and horizontal distance between the two end pixels $(x_1, y_1)$ and $(x_2, y_2)$ of the major axis, and S is the area of the object (i.e., the number of white pixels in the object).

The orientation angle of the object was defined as

$$\varphi = \arctan\left(\frac{\delta y}{\delta x}\right). \tag{2}$$

The aspect ratio and orientation angle were calculated for each extracted image of eel or stick/pipe. The distinct statistical attributes for eels and sticks/pipes in terms of shapes and orientation in the images serve as a basis for the use of more complex classification approaches to distinguish them.

2.2.5. Convolutional Neural Network

In this study, the CNN architecture (Figure 5) was composed of the input layer, convolutional layer, max-pooling layer, fully connected layer, and output layer. The first

convolutional layer had 32 filters with a filter size of $5 \times 5$. The second convolutional layer had 64 filters with the same filter size. A max-pooling layer followed with a $2 \times 2$ pooling size. Before and after the fully-connected layer with 128 hidden units, dropouts were implemented to prevent over-training and generalize better [37]. The activation function was the rectified linear unit (ReLU). The output layer used the sigmoid function as the final classifier. The weights were updated using the back-propagation algorithm [38]. Adam was chosen as the optimizer [39]. Binary cross-entropy was used as the loss function,

$$J = -\frac{1}{m} \sum_{j=1}^{m} \left[ y^{(j)} \log\left(\hat{y}^{(j)}\right) + \left(1 - y^{(j)}\right) \log\left(1 - \hat{y}^{(j)}\right) \right], \tag{3}$$

where $\hat{y}$ the output of the model; $y$ was the true label of the input sample; $j$ was the sample index; and $m$ was the size of the training data.

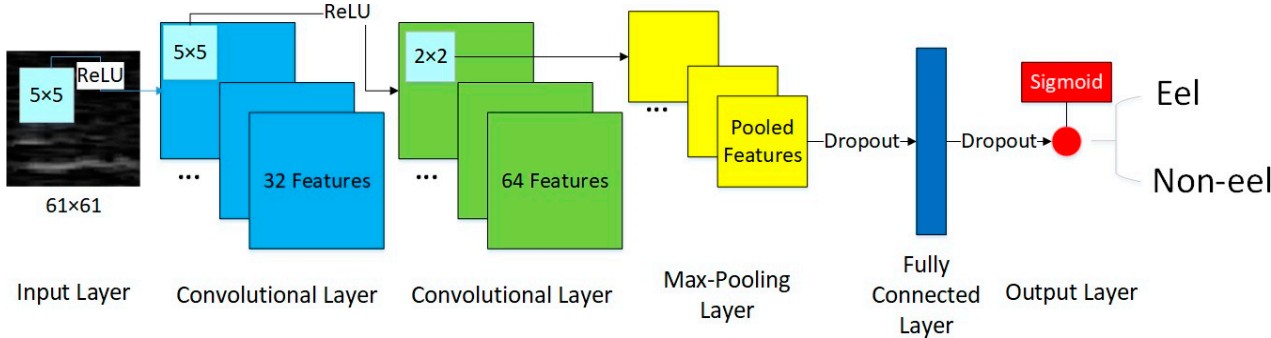

**Figure 5.** Architecture of the convolutional neural network for the classification of extracted sonar images of eels and non-eel objects.

The CNN model was built using Tensorflow 1.0 and trained and tested on the laboratory data and the field data, respectively. Each dataset (set of video clips) was split into the training set (~80%) and the testing set (~20%). The training set was used to train the weights of the network. The testing set was never seen by the model in the training stage and was used to evaluate the performance of the trained CNN model. Due to the differences in the sample size of the laboratory and field data, a different method was employed to assess the performance of CNN of each dataset. For the laboratory data, ten-fold cross validation was used to evaluate the classification accuracy in an image-based manner. For the field data, the classification accuracy was calculated in a video-based manner, with each video clip being classified by the ensemble classification results of the images extracted from the clip. Because the images in the same video clip might be classified into different groups, a threshold is needed for the percentage of images classified as eel by the CNN to make a video-based classification decision. If the percentage exceeds the threshold, then the tested video clip is classified as eel; otherwise, the video clip is classified as non-eel. When an object moves through the sonar view, it is recorded as a series of images in a video clip. The target should be classified based on the ensemble classification results of all relevant images as they belong to the same target. This design aligns with the working scheme in field operations and is consistent with the decision-making process of biologists and technicians trained to identify targets from sonar images.

## 3. Results

### 3.1. Statistical Analysis of Sonar Images: Aspect Ratio and Orientation Angle of Targets in the Laboratory

The aspect ratio and orientation angle of eels and sticks are shown in Figure 6, with the results of individual targets in the Supplementary Materials (Figures S4 and S5). The summary statistics of aspect ratio and orientation angle illustrate clear differences between eels and neutrally buoyant sticks in sonar images. Eels have more variability than sticks

in both aspect ratio and orientation angle. The observations were consistent with our assumption that non-eel objects usually have a more rigid shape than eels, since the body of eels can twist freely (anguilliform swimming motion), while eels often have a 'freestyle' component in their movements. Overall, eels have a more variable shape than sticks, and aspect ratio and orientation angle can be qualitatively used to differentiate eels from sticks. This analysis gives us the confidence that eels and sticks can be distinguished in sonar images.

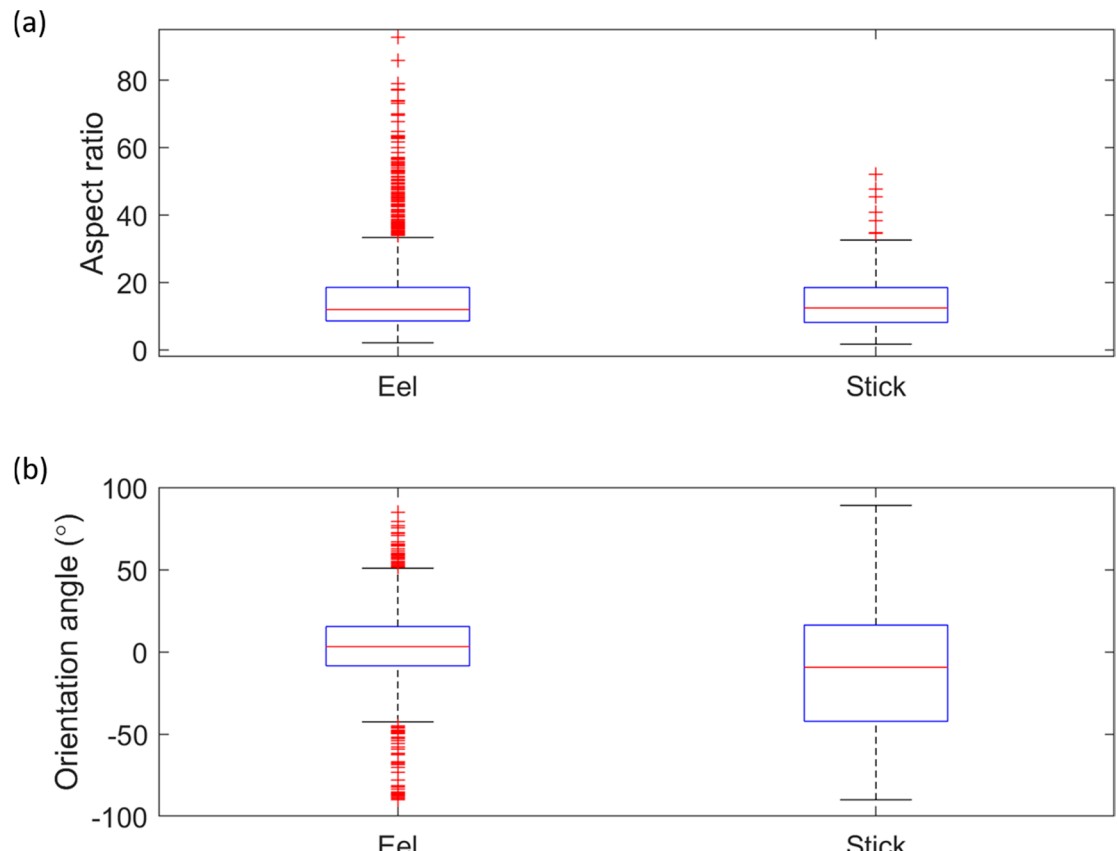

**Figure 6.** The aspect ratio (**a**) and orientation angle (**b**) of eel and stick in the laboratory experiments.

### 3.2. CNN Performance Evaluation with the Laboratory Data

The number of extracted images for eels and sticks are listed in Table 2. In total, there were 1892 eel images and 3386 stick images. Each image had four versions: (1) original image without any pre-processing; (2) image denoised with wavelet transform; (3) image processed with background removal; and (4) image processed by both background removal and wavelet denoising. The classification accuracies based on different versions of images were first evaluated. For each version of images, ten-fold cross-validation was used to assess the performance of the CNN.

**Table 2.** Number of extracted images of eels and sticks in the laboratory experiments.

| Object ID | Water Flow Speed (m/s) | Number of Images |
|---|---|---|
| Eel_1 | 0.76 | 124 |
| Eel_1 | 0.53 | 110 |
| Eel_2 | 0.76 | 36 |
| Eel_2 | 0.53 | 360 |
| Eel_3 | 0.76 | 686 |
| Eel_3 | 0.53 | 328 |
| Eel_4 | 0.76 | 222 |
| Eel_4 | 0.53 | 26 |
| Stick_1 | 0.76 | 785 |
| Stick_1 | 0.53 | 869 |
| Stick_2 | 0.76 | 972 |
| Stick_2 | 0.53 | 760 |

Table 3 shows the overall classification accuracies of the four versions of images. Data with both background removal and wavelet denoising yielded the best results, with approximately 1% increase in accuracy compared to the other types of data, which indicates that background removal and wavelet denoising improved the classification accuracy. In addition, the classification accuracy at the low flow speed was higher than that at the high flow speed, which means that the low flow speed is more favorable for eel identification than the high flow speed in the current experimental design.

**Table 3.** Image-based overall accuracies of CNN classification of the laboratory sonar images.

| Water Flow Speed | Image Processing | Image-Based Accuracy |
|---|---|---|
| Two flow speeds (0.76 m/s and 0.53 m/s) | Original | 97.33% $\pm$ 1.78% |
|  | Wavelet denoising only | 97.65% $\pm$ 1.74% |
|  | Background subtraction only | 97.62% $\pm$ 1.61% |
|  | Background subtraction and wavelet denoising | 98.42% $\pm$ 1.29% |
| High flow (0.76 m/s) | Background subtraction and wavelet denoising | 97.88% $\pm$ 2.30% |
| Low flow (0.53 m/s) | Background subtraction and wavelet denoising | 99.15% $\pm$ 1.30% |

### 3.3. CNN Performance Evaluation with the Field Data

The field datasets had 13 eel clips and 15 stick/pipe clips with a total of 301 and 252 extracted images, respectively. The number of extracted images and the sonar setting parameters of each clip are listed in Tables S1 and S2 in the Supplementary Materials. The eel clips were graded into two tiers: clips in Tier 1 showed clear sinusoidal locomotion in at least five frames, while clips in Tier 2 showed clear sinusoidal locomotion in at least two frames of the clip, based on professional judgement. The CNN model was trained on 10 eel clips and 12 stick/pipe clips, and then tested on three eel clips and three stick/pipe clips. The training data and testing data were separated by randomization. The averaged results of 50 randomizations were calculated to assess the false negative rate and false positive rate (Figure 7). When the threshold of percentage of images in the video clip increases, the false positive rate decreases, and the false negative rate increases accordingly. When the threshold was set to 70.0%, the false positive rate was 9.3% and the false negative rate was 13.3%. When the threshold was set to 60.0%, the false positive rate was 12.0% and the false negative rate was 10.0%. Note that these results represent the classification results of the video clips. The explored hyperparameters and the optimal hyperparameters are listed in Table 4.

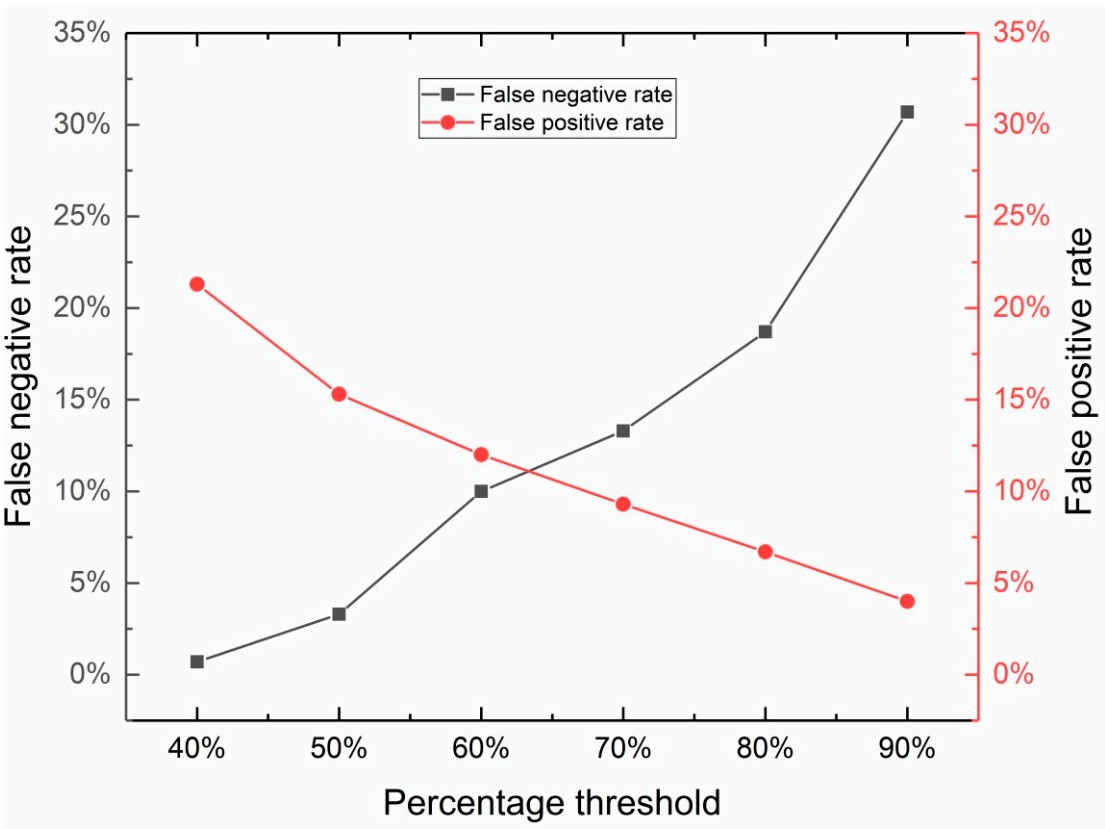

**Figure 7.** The classification results of the sonar video clips from the field experiments. The percent-age threshold was set for the ratio of the number of images classified as eels by the CNN to the total number of extracted images in the testing sonar video clip.

**Table 4.** The explored hyperparameters and the optimal hyperparameters of the CNN model trained and tested on the field data.

| Hyperparameters | Explored Values | Optimal Values |
| --- | --- | --- |
| Batch size | 16, 32 | 32 |
| Number of epochs | 4, 5, 6, 7, 8 | 5 |
| Learning rate | 0.00005, 0.0001, 0.001 | 0.0001 |
| Weights | 0.4 and 0.6; 0.5 and 0.5; 0.6 and 0.4 | 0.4 and 0.6 |
| Training vs. testing split | 80% and 20%; 70% and 30%; 60% and 40% | 80% and 20% |

*3.4. Transferability from the Laboratory Data to the Field Data*

We explored the transferability of the model from the laboratory data to field data. Due to the scarcity of existing field data and the high costs of performing field tests, we made full use of the laboratory data and tested the applicability of the model trained on the laboratory data to the field data. It was found that when training on the laboratory data and testing on the field data (with the six eel clips in Tier 1 and one stick clip), the image-based accuracy was only 50.4%, indicating that the model trained on the laboratory data cannot be directly used for the field cases. Then, we mixed the laboratory and some of the field data for training and the remaining field data for testing. For example, we chose three out of the six field clips (from Tier 1) to be switched to the training set. In this case, the image-based overall accuracy increased to 89.6%, and the video-based classification reached a zero false positive rate and zero false negative rate when the threshold was 70.0%.

**4. Discussion**

The classification results from the laboratory data successfully demonstrated the validity of CNN for sonar image classification. It was also shown that the preprocessing of

sonar images with background removal and wavelet denoising could enhance the sonar images and improve classification accuracies. As expected, the results from the field data were not as good as those from the laboratory data due to the differences in noise levels, eel behaviors in different environments, and different detection ranges. Very low levels of noise were observed in the background of the sonar images due to the controlled conditions in the laboratory. In comparison, the field images were much noisier, and the eel behaviors were also different. In the laboratory tank, eels were tethered to swim against the flow in a limited space, and the flow field generated by the pump arrays were occasionally turbulent. Eels were observed struggling with the flow in order to maintain body control from time to time. The field experiments had a more steady flow, and the eels were mostly swimming in a normal sinusoidal locomotion pattern. Since eels had complete body control in the field, their body orientation relative to the sonar beam axis was more variable compared to the tethered eels in the laboratory. Thus, the reflection of sound signals and the body shape of eels in projected sonar images were more variable in the field than in the laboratory. In addition, since water flow speeds in the field were relatively high, eels remained within the acoustic field for a relatively short duration of time, thereby limiting the number of images available for analysis. Finally, the detection range of the field data (10–35 m) was longer than the laboratory data (~6 m) (i.e., the resolution of sonar images in the field was lower than in the laboratory). All these factors contributed to the complexity of sonar images in the field, and thus increased the difficulty of image classification by CNN.

Regardless of all the challenges in the field data, the accuracies achieved by the CNN were comparable to that of human experts. It was reported by EPRI [12] that the classification test achieved 0% false positive with a 20% false negative error rate and a 12% false positive with 0% false negative error rate depending upon the choice of qualitative threshold classification criterion. Note that the sample size in the EPRI study [12] was smaller than in this study. With more data added to the training set, it is expected that the CNN model can outperform human experts in sonar image classification accuracy; additionally, the CNN model supports much higher levels of automation and efficiency compared to human-supervised classification.

The conditions favorable for sonar image classification by CNN were also assessed. The laboratory experiments showed that at the low flow speed, the accuracy was higher than at the high flow speed. This result can be attributed to the eel's body control under different flow speeds. When eels were swimming against high flow, it was hard for them to maintain body control or swim in the sinusoidal locomotion pattern. It should be noted that the values of high and low water flow speeds were not exactly equivalent to the flow speeds in the field, where eels were not tethered and could swim with or against the flow or did not swim at all. The effect of natural river flow on CNN classification accuracies should be further investigated through a set of field tests with measured flow conditions.

The transferability of laboratory data to the field was briefly addressed. It was noted that some aspects of eel behavior in the laboratory differed from that observed in the field. To further explore the transferability of laboratory derived data and models to the field, transfer learning should be investigated [40].

In the field tests, the detection range of the sonar varied from 10 to 35 m. The size and shape of the same target appeared differently in sonar images depending upon the range. In addition, the targets' signal strengths varied at different range. The difference in pixel intensities can be handled by image normalization. Moreover, the CNN model can handle these scaling and pixel intensity difference issues well. For example, in the 50 randomizations of the field data classification, the two eel clips with 35 m detection range were split into testing data 10 and 12 times, respectively, and they were always correctly classified as eels by CNN.

It should be noted that the classification in this study was done based on static images extracted from sonar videos, and not on the video clips with cross-frame information. The motion of targets between sonar frames is a main feature that human experts use to identify eel and non-eel objects. Incorporating motion analysis in future work would be helpful to

improve object detection accuracy. Moreover, in future work, additional field or laboratory data will be integrated to build a more robust model that can achieve high accuracies under various riverine environments. Furthermore, the non-eel targets investigated in this study only covered water-logged sticks and PVC pipes, so the future work should investigate other non-eel objects such as non-eel fish species and macrophyte mats.

## 5. Conclusions

This study successfully identified American eels from sonar images using deep learning. Results from the laboratory experiment data showed that the analytics of background subtraction and wavelet denoising enhanced sonar images and increased CNN classification accuracies. The CNN classification accuracy of field data was commensurate with that achieved by human experts. The designed sonar image processing and classification method will enable the automation of fish identification. With potential applications to fish monitoring near hydropower projects, this method will facilitate research and development related to eel passage at hydropower facilities and increase the efficiency of hydropower operations while preserving a friendly environment for fish passage and migration.

**Supplementary Materials:** The following are available online at https://www.mdpi.com/article/10.3390/rs13142671/s1, Figure S1: Tethering of the eel and sticks in the laboratory experiments, Figure S2: Photo of the pump array, the overhead camera (circled), and the anechoic barrier, Figure S3: Field experiment setup, Figure S4: The aspect ratio and orientation angle of eels in the laboratory experiments, Figure S5: The aspect ratio and orientation angle of sticks in the laboratory experiments, Table S1: Sonar settings and extracted images of eel data from the field experiments, Table S2: Sonar settings and extracted images of sticks and PVC pipes from the field experiments.

**Author Contributions:** Z.H., Z.D.D., and P.T.J. conceived the ideas and designed methodology; R.P.M. and X.Z. collected the laboratory data; P.T.J. and Z.D.D. directed the study; P.T.J. curated the field data; X.Z., T.Y., and Z.H. analyzed the data; X.Z. led the writing of the manuscript. All authors contributed critically to the drafts and gave final approval for publication. All authors have read and agreed to the published version of the manuscript.

**Funding:** This study was funded by the U.S. Department of Energy Water Power Technologies Office and the Electric Power Research Institute with Award No. (DE-EE0008341).

**Institutional Review Board Statement:** The PNNL animal facilities are certified by American Association for Accreditation of Laboratory Animal Care. Fish were handled in accordance with federal guidelines for the care and use of laboratory animals, and study protocols were approved by PNNL Institutional Animal Care and Use Committee protocol code 2018-12 on 11/05/2018.

**Informed Consent Statement:** Not applicable.

**Data Availability Statement:** Data from both laboratory and field experiments can be found through https://datadryad.org/stash/share/s6otq-YuRjJsCnkHq_wtjPk08DM_UPORfzDNpmuwv_k (accessed on 06/30/2021).

**Acknowledgments:** The authors would like to thank Bill Hanot at Sound Metrics Corp. for the loan of the ARIS Explorer 1800 sonar for the laboratory tests.

**Conflicts of Interest:** The authors declare no conflict of interest.

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
