# Peer review of "Deep Learning for Automated Detection and Identification of Migrating American Eel Anguilla rostrata from Imaging Sonar Data"

_remotesensing, doi:10.3390/rs13142671_

Round 1
Reviewer 1 Report
- The language style used in the article is good, where most of the explanations are clear.
- Minor issue in Figure 2b - in the figure description, it is mentioned that subtraction was done from 4 adjacent frames. But on page-6 line 220, the authors mention 6 adjacent images.
Reviewer 2 Report
The authors are describing well given problem. For me personally it was very educational research article. I would like to propose some adjustments, which would improve overall quality of the research paper:
- Figure 1. Has black part of image in which Eel and other things should be visible. Unfortunately, these are not visible due the low contrast. Please make this image more visible.
- It is not clear what kind of platform was used for CNN training.
- References consist of very old references, which are dated from 1989. A novel research work should consist of references not older than 5-8 years. Please fix that.
- Authors should explain why they are not using novel object detection algorithms such as FASTER RCNN, YOLO, SSD.
- UNet type deep neural network is perfect for images such as sonar or xray images. Why authors do not use this one instead of CNN?
Reviewer 3 Report
I wonder if the authors can compare their CNN model with other SOTA cnn models.
Reviewer 4 Report
A well written paper. Just minor comments:
On line 62: please explain what ARIS is, when the instrument is mentioned for the first time.
It is really difficult to distinguish any details in the grayscale images (Fig 1 and Fig 2) , would it be possible to change the output LUT to enhance the low intensity pixels a bit more?
Reviewer 5 Report
- In the title, why "Anguilla rostrata" is italic?
- It would be better if the "related works" section was separated.
- In fig.2, "The x and y labels denote the pixel numbers" but labels are not in the figures.
- "Since only one object of interest was recorded in each video clip after data cleaning," what do you mean by "data cleaning"?
- "A threshold of pixel intensity was selected," how? how did you find a proper threshold? What is the threshold value?
- "The extracted objects were visually examined to ensure that they were the intended targets" here "visually examined" what do you mean by that? Did you decide if it is a target by manually with your own eyes?
- Fig.3 has been moved to the right and the part of it is not visible.
- In the field experiment, when you collect data of eels for training CNN, what sensor did you use to know if there was an eel under water? Because you do not know there are eels until you put them under water yourself or directly check by eyes. If you used the same sensor that you used in the lab, then what was the distance between eels and the sensor? I wander how you knew the collected data from the field were exactly data of eels. Can you check eels by own eyes when they are under water?
- What sticks did you use? how they look like? Do they look similar to eels? What is the size of eels and sticks? Need example images of eels and sticks you used in the experiments. Also dimensions and weights of the eels and sticks need to me included.
- When you say a stick, I imagine that it is a wooden stick. If it is a wooden stick then it wont sink under water but eels are under water. If the eel you are saying is a sort of fish, then why do you compare a living animal with non living wooden stick? I think similar sorts of fish should be compared with the eels (more than 3 sorts of fish).
- Is there only eels and sticks in that river (Iroquois Dam looks very big)? Or there are another fishes? Classifying a single type of living fish with a stick is not challenging then the study is not acceptable. Or the importance of classifying eels and sticks should be detailed.
- Why did you use the filter size of 5×5?
- Structure of CNN should be detailed including size of feature maps, description of hyperparameters and trainable parameters.
- There are common measurements for identification problem, authors should measure the results using the measurements.
- If the study was about to classify similar sorts of fishes, then it is an identification problem. However, it is just to classify a fish and a stick then it is just a classification or recognition problem.
Reviewer 6 Report
This study successfully identified American eels from sonar images with deep learning. Results from the laboratory experiment and field data showed that the background subtraction and wavelet denoising enhanced sonar images and increased CNN classification accuracy.
Comments to the authors:
- At the end of Introduction, the author said Sea lamprey and Pacific lamprey have comparable body shape and swimming behavior to eel, so whether the method proposed in this paper can be used for anti-interference of these two kinds of fish
- In Figure 2 on page 6, how many pictures are used in the background subtraction? Four adjacent frames are written in the caption(b) (line 205), but six adjacent frames are written on line 220.
- What do the 6 images in each row of Figure 3 and Figure 4 represent? Are they images of different angles of a target or represent different targets?
- In Figure 7 on page, when the threshold of percentage of images in the video clip increases, the false positive rate decreases and the false negative rate increases accordingly. This should be common sense, what the author's intention is?
- On page 14 line 457, I can not find the Figure S1~S5 in this paper.
- It is recommend the authors to compare with other methods to show the innovation of the proposed method.
Round 2
Reviewer 3 Report
The revised version can be accepted.
Reviewer 5 Report
- to answer 5, "The threshold of pixel intensity was selected through trial and error. For the laboratory data, the selected threshold value was 30. For the field experiment data, the selected threshold value was 25." Have you included this explanation about threshold 25 and 30 in the paper? i could not find those threshold numbers in the edited version.
- I suggest the authors to include some of important review responses (explanations to the comments) in the discussion section.
- to response 15, as outputs (as shown in figure 5) are recognized as eels and non-eels, it is binary classification problem. In detail, the proposed method detects an object (an eel or a stick) then recognize if it is an eel. Thus, i suggest authors to edit the title and contents, and remove the words "identification or identify" and use "binary classification" and "recognition".
Reviewer 6 Report
Weak innovation is not enough to publish.